

# The BErkeley Atmospheric CO$_2$ Observation Network: Field Calibration and Evaluation of Low-cost Air Quality Sensors

Jinsol Kim[1], Alexis A. Shusterman[2], Kaitlyn J. Lieschke[2], Catherine Newman[2], and Ronald C. Cohen[1,2]

[1]Department of Earth and Planetary Science, University of California Berkeley, Berkeley, CA 94720, USA
[2]Department of Chemistry, University of California Berkeley, Berkeley, CA 94720, USA

*Correspondence to*: Ronald C. Cohen (rccohen@berkeley.edu)

**Abstract.** The newest generation of air quality sensors is small, low cost, and easy to deploy. These sensors are an attractive option for developing dense observation networks in support of regulatory activities and scientific research. They are also of interest for use by individuals to characterize their home environment and for citizen science. However, these sensors are
difficult to interpret. Although some have an approximately linear response to the target analyte, that response may vary with time, temperature, and/or humidity, and the cross-sensitivity to non-target analytes can be large enough to be confounding. Standard approaches to calibration that are sufficient to account for these variations require a quantity of equipment and labor that negates the attractiveness of the sensors' low cost. Here we describe a novel calibration strategy for a set of sensors including CO, NO, NO$_2$, and O$_3$ that makes use of multiple co-located sensors, a priori knowledge about the
chemistry of NO, NO$_2$, and O$_3$, as well as an estimate of mean emission factors for CO and the global background of CO. The strategy requires one or more well calibrated anchor points within the network domain, but it does not require direct calibration of any of the individual low-cost sensors. The procedure nonetheless accounts for temperature and drift, in both the sensitivity and zero offset. We demonstrate this calibration on a subset of the sensors comprising BEACO$_2$N, a distributed network of approximately 50 sensor "nodes," each measuring CO$_2$, CO, NO, NO$_2$, O$_3$ and particle matter at 10
second time resolution at approximately 2km spacing in locations surrounding the San Francisco Bay Area.

## 1 Introduction

In urban environments, air quality has complex spatial and temporal patterns. Diverse emission sources are present with large variations in emission rate and source type on scales of hundreds of meters. In addition, dispersion of pollutants into the urban environment is affected by the topography of the urban landscape and the associated wind flows, which also vary
on length scales of ~100 m (Vardoulakis et al., 2003; Lateb et al., 2016). Conventional approaches to air quality monitoring rely on a limited number of relatively high cost instruments that lack the spatial resolution needed to characterize these variations, opting instead to target spatial averages. This averaging hampers our attempts at source attribution and understanding of mixing, chemistry, and human exposure in cities where emissions vary on spatial scales that are small compared to typical observations or models.



One approach to obtaining higher spatial resolution in observations is passive sampling, which has been implemented as a low-cost method using inexpensive sampling devices that can be later analyzed in bulk. Passive samplers do not require electrical power to function properly and are collected and analyzed one to two weeks after deployment. Such protocols

provide high spatial resolution but also have significant drawbacks. Spatial resolution is gained at the expense of temporal resolution, and analysis after collection of the samplers is time consuming, thus passive sampling has typically been used only in short duration experiments (e.g. Krupa & Legge, 2000; Cox, 2003). Furthermore, as a result of boundary layer dynamics, passive sampling in urban areas is likely dominated by the high concentrations found at night and relatively insensitive to daytime variability.

Recent developments in low-cost sensors for trace gases and particulate matter, as well as advances in software and hardware enabling low-cost data communication, have made high-density air quality monitoring networks possible. Devices and networks of devices are emerging that are low cost, report at high time resolution, and are capable of long-term deployment, providing potential for improvement over the two major weaknesses of the passive sampling. Examples include

metal oxide sensors used to measure $O_3$, CO, $NO_2$, and total VOCs (Williams et al., 2013; Bart et al., 2014; Piedrahita et al., 2014; Moltchanov et al., 2015; Sadighi et al., 2017), and electrochemical sensors used to measure CO, NO, $NO_2$, $O_3$, and $SO_2$ (Mead et al., 2013; Sun et al., 2015; Jiao et al., 2016; Hagan et al., 2017; Jerrett et al., 2017; Michael et al., 2017). These different low-cost sensor systems were compared during the 1[st] EuNetAir Air Quality Joint Intercomparison Exercise in 2014 (Borrego et al., 2016). While these studies found low-cost trace gas sensors to be successful at qualitatively

characterizing the variability of air quality in an urban area, challenges related to selectivity and stability remain, hindering more quantitative interpretation of the data.

The current generation of low-cost sensors is not as easily tied to a gravimetric calibration standard as many of the passive samplers. Calibration is known to vary with sensor age, temperature, and in some cases humidity. In addition, many of the

sensors have responses to gases other than the target analyte (Mead et al., 2013; Spinelle et al., 2015; Cross et al., 2017; Michael et al., 2017; Mijling et al., 2017; Spinelle et al., 2017; Zimmerman et al., 2017). One approach to addressing this challenge is to combine periodic re-calibration and co-location with regulatory reference instruments in the lab or the field (Williams et al., 2013; Moltchanov et al., 2015; Jiao et al., 2016; Mijling et al., 2017). Field calibration is preferred as in-lab performance is often a poor approximation of sensor behavior under ambient conditions (Piedrahita et al., 2014; Masson et

al., 2015). However, either method requires considerable time investment by trained personnel, especially as the number of sensors increases. The requirement of time and labor consuming calibration then offsets the low-cost advantage of the sensors.



In this paper, we explore an automated, in situ strategy for the calibration of individual sensors embedded in an air quality sensor network that includes both low-cost sensors and anchor points of higher grade, well calibrated instrumentation. The BErkeley Atmospheric $CO_2$ Observation Network (BEACO$_2$N) is a low-cost, high-density greenhouse gas ($CO_2$) and air quality (CO, NO, $NO_2$, $O_3$, and particulate matter) monitoring network located in San Francisco Bay Area, California (see

Fig. 1 and Shusterman et al. 2016). As of this writing, BEACO$_2$N consists of approximately 50 sensor "nodes," deployed with approximately 2 km horizontal spacing. Most of the nodes are mounted on the roofs of schools and museums. In previous work, we described an approach to $CO_2$ sensing and calibration (Shusterman et al. 2016). Here, we focus on CO, NO, $NO_2$, and $O_3$.

We begin by describing laboratory experiments and in-field comparisons to co-located reference instruments that give an initial characterization of the sensors and provide insight into the effects of temperature, humidity, and cross-sensitivity to non-target analytes. Then we describe an in situ calibration procedure that accounts for these variables without requiring co-location with a reference instrument. The calibration procedure is finally verified against regulatory quality measurements not used in the procedure itself.

## 2 Instrument Description

Details of the node design and deployment are described in Shusterman et al. (2016). Briefly, each BEACO$_2$N node contains a Vaisala CarboCap GMP343 non-dispersive infrared sensor for $CO_2$, a Shinyei PPD42NS nephelometric particulate matter sensor, and a suite of Alphasense electrochemical sensors: CO-B4, NO-B4, either $NO_2$-B42F or $NO_2$-B43F, and either $O_x$-B421 or $O_x$-B431. All sensors are assembled into compact, weatherproof enclosures as shown in Fig. 2. Two 30 mm fans are

located on either side of the enclosure to facilitate airflow through the node. A Raspberry Pi microprocessor collects data via a serial-to-USB converter for $CO_2$ and an Adafruit Metro Mini microcontroller for all other sensors. Then, data collected every 5 or 10 seconds is transmitted to a central server using a direct on-site Ethernet connection, a local Wi-Fi network, or an Adafruit FONA MiniGSM cellular module.

The Alphasense B4 electrochemical gas sensing series that we use employs a four-electrode approach. The electrodes are embedded in an electrolyte solution separated from the atmosphere by a semi-permeable membrane. The gas of interest diffuses through the membrane into the electrolyte where it contacts a "working" electrode, and is either oxidized (in the case of NO and CO) or reduced ($NO_2$ and $O_3$). The potential at the working electrode is maintained at a constant value with respect to a "reference" electrode. Electric charge produced at the working electrode is balanced by the complementary

redox reaction at a "counter" electrode, generating an electric current. The sensor also contains an "auxiliary" electrode, which shares the working electrode's catalyst structure, but is isolated from the ambient environment, accounting for



fluctuations in the background current associated with other processes at the electrode and electrolyte. Subtracting the auxiliary current from the working current gives a corrected current dependent on the gas concentration.

The current detected by the sensors is converted to a voltage that is related to gas concentration using amplifiers in the Individual Sensor Boards (ISBs) provided by Alphasense. Over the mixing ratio range of interest, the sensors' responses to the gases of interest are approximately linear. We derive mixing ratio from the observed voltages by subtracting an offset and then scaling by a constant (Eqn. 1-4):

$$CO_{ambient} = (V_{CO} - zero_{CO})/k_{CO} \tag{1}$$

$$NO_{ambient} = (V_{NO} - zero_{NO})/k_{NO} \tag{2}$$

$$NO2_{ambient} = (V_{NO2} - zero_{NO2})/k_{NO2} - r_{NO-NO2} \times NO_{ambient} \tag{3}$$

$$O3_{ambient} = (V_{O3} - zero_{O3})/k_{O3} - r_{NO2-O3} \times NO2_{ambient} \tag{4}$$

Here, CO, NO, $NO_2$, and $O_3$ with the subscript "ambient" refer to the gas mixing ratios (ppb) in air; $V_{CO}$, $V_{NO}$, $V_{NO2}$ and $V_{O3}$ are the signals (mV) measured by each sensor; $zero_{CO}$, $zero_{NO}$, $zero_{NO2}$ and $zero_{O3}$ indicate the voltage measured in the absence of analyte; and $k_{CO}$, $k_{NO}$, $k_{NO2}$ and $k_{O3}$ represent the linear sensitivity factor that converts mV to ppb. Additional terms corresponding to the cross-sensitivities of the $NO_2$ and $O_3$ sensors appear in Eqn. 3 and 4, where $r_{NO-NO2}$ is the cross-sensitivity of the $NO_2$ sensor to NO gas and $r_{NO2-O3}$ is the cross-sensitivity of the $O_3$ sensor to $NO_2$ gas.

There are a total of 8 sensitivities and zero offsets, as well as 2 cross-sensitivity terms. All of these may also vary with time, temperature, and humidity. Thus we need a calibration strategy that constrains 10 parameters in a single instant as well as the variation of those 10 parameters in response to the environmental variables. We begin by characterizing the sensors in both laboratory and outdoor environments.

We evaluate BEACO$_2$N in terms of four factors: drift, noise, cross-sensitivity, and temperature dependence. The humidity dependence is included in the temperature dependence, as there is no evidence for independent humidity dependence and relative humidity exhibits an anti-correlation with temperature in the field. This paper examines the behavior of CO-B4, NO-B4, $NO_2$-B42F, and $O_x$-B421. The more recently released $NO_2$-B43F and $O_x$-B431 sensors respond differently; their performance will be assessed in a future study. In the laboratory, a range of mixing ratios of target gases were delivered to a chamber containing the full suite of four Alphasense B4 sensors: CO, NO, $NO_2$, and $O_3$. Zero air was supplied by a Sabio 1001 Compressed Zero Air Source and blended with calibration gases using a ThermoScientific 146i Multi-Gas Calibrator.



*Noise* – Alphasense reports $2\sigma$ noise of $\pm 4$ ppb, $\pm 15$ ppb, $\pm 12$ ppb, and $\pm 15$ ppb for CO, NO, $NO_2$, and $O_3$, respectively over concentrations from 0 ppb to 200 ppb. In our laboratory, noise ($\pm 2\sigma$) was measured for ambient ppb levels and was seen to be $\pm 8$ ppb for CO, $\pm 4$ ppb for NO, $\pm 5$ ppb for $NO_2$, and $\pm 11$ ppb for $O_3$.

*Cross-Sensitivity* – We measured the cross-sensitivity of all 4 of the trace gas sensors to the non-target gases. The $NO_2$ sensors (NO2-B42F) and $O_3$ sensors (Ox-B421) were the only ones to exhibit sensitivity to other species. The $O_3$ sensor demonstrated 100% sensitivity to $NO_2$. This sensor is now being marketed by Alphasense as an odd oxygen ($O_x \equiv O_3 + NO_2$) sensor. In addition, the $NO_2$ sensor was found to possess a significant NO sensitivity (130%) that exceeds the cross-sensitivity specified in the Alphasense documentation (<50%). These cross-sensitivities are represented in Eqn. 3 and Eqn. 4.

   *Temperature Dependence* – Electrochemical sensors are known to have temperature dependent sensitivities and zero offsets. Alphasense reports sensitivities and zero offsets for a temperature range between -30 °C and 50 °C. The sensitivities in their data sheets vary with temperature by +0.1 to +0.3 %/K and the zero offsets are indicated to vary little except at high temperatures. We observed similar, but slightly larger variations via in situ comparison to co-located reference instruments.

We observed temperature dependence in the sensitivities of +0.3 to +5 %/K and no variation in the zero offset of the CO, $NO_2$, and $O_3$ sensors from 10 °C to 24 °C (Fig. 3). However, the zero offset of the NO sensor exhibited a strong temperature dependence of 0.34 mV/K.

   *Drift* – Two laboratory calibrations were performed roughly 10 weeks apart and the zero offsets and sensitivities are shown
in Table 1. Over the 10-week interval, zero drift was equivalent to -15.9 ppb, -2.3 ppb, +15.8 ppb, and -12.7 ppb for CO, NO, $NO_2$, and $O_3$, respectively. Alphasense reports the stability over time for the zero offset to be < $\pm 100$, 0 to 50, 0 to 20, and 0 to 20 ppb $yr^{-1}$ for these sensors, respectively; over this 10 week interval, the observed zero drift was within the range of these specifications. However, it is a large fraction of the annual drift specification and further experiments would be warranted to test whether the zero measured is stable over a full year within the specified tolerances. The drift in the
sensitivity (in % of $k_X$) was -15.9%, -17.7%, -20.6%, and -53.2%. Alphasense reports <10, 0 to -20, -20 to -40, and < -20 to -40% $yr^{-1}$ for CO, NO, $NO_2$, and $O_3$ calibration factors, respectively. We find that drift for the CO and $O_3$ sensitivities exceeded the manufacturer specifications, but that the NO and $NO_2$ sensitivity drifts were within the specified tolerances.

### 3 Model for Field Calibration

   Here, we propose a model for field calibration that leverages (1) useful cross-sensitivities, (2) chemical conservation
equations, (3) knowledge of the global and/or regional background of pollutants, and (4) assumptions based on well-known characteristics of urban air quality and local emissions. The result is a calibration procedure for the drift and temperature dependencies of the 10 calibration parameters that does not require co-location with a reference instrument or prior



laboratory experiments for each sensor. The first constraint we apply is the $O_3$ sensors' cross-sensitivity to $NO_2$. Laboratory measurements indicate that this cross-sensitivity is 100% and we fix it at that value.

### 3.1 Regional ozone uniformity to calibrate the NO, NO$_2$ and O$_3$ sensors' sensitivities

The NO, NO$_2$, and O$_3$ sensitivity can be derived from observations with higher quality instruments at nearby locations.

Ozone is a secondary pollutant with small local scale variation, except in the very near field of NO emissions. The Bay Area Air Quality Management District (BAAQMD) maintains four TECO 49i ozone analyzers within the BEACO$_2$N study area (see Fig. 1). We choose the closest site among these four regulatory monitoring sites to provide $O3_{ambient}$ as a constraint for multiple linear regression of Eqn. 5 (derived from Eqn. 2-4).

$$O3_{ambient} = \frac{V_{O3}}{k_{O3}} - \frac{V_{NO2}}{k_{NO2}} + r_{NO-NO2}\frac{V_{NO}}{k_{NO}} - offset \tag{5}$$

Here, $offset$ is a combination of the zero offsets of the NO, NO$_2$, and O$_3$ sensors, all of which can be constrained as detailed in Sect. 3.2 below. The sensitivity of O$_3$ and NO$_2$ sensors ($k_{O3}$ and $k_{NO2}$), and relationship between the NO-NO$_2$ cross-sensitivity and the sensitivity of the NO sensor ($r_{NO-NO2}/k_{NO}$) are obtained by doing multiple linear regression of Eqn. 5.

### 3.2 Use of chemical conservation equations near emissions to calibrate the NO, NO$_2$ and O$_3$ sensors' sensitivities and zero offsets

We are able to constrain the sensitivity of NO sensors by taking advantage of proximity to local emission sources and the following chemical conservation equations.

$$NO + O_3 \rightarrow NO_2 + O_2 \tag{R1}$$

$$NO_2 + h\nu \rightarrow NO + O \tag{R2}$$

$$O + O_2 + M \rightarrow O_3 + M \tag{R3}$$

These three reactions result in a steady-state relationship among the nitrogen oxides ($NO_x \equiv NO + NO_2$) and ozone. This steady-state operates on a time scale of approximately 100 seconds. On these short time scales, loss of NO (or NO$_2$) and production of NO$_2$ (or NO) is equal and the sum of NO and NO$_2$ is conserved. Similarly, O$_x$, the sum of NO$_2$ and O$_3$ is conserved. Thus we expect that increases (decreases) in NO and O$_3$ are exactly balanced by corresponding decreases (increases) in NO$_2$. Properly calibrated time derivatives of NO vs. time derivatives of NO$_2$ and derivatives of NO$_2$ vs. time

derivatives of O$_3$ will have a slope of -1 (see Fig. 4), which can be expressed as Eqn. 6:

$$\frac{\Delta NO_{ambient}}{\Delta NO2_{ambient}} = \frac{\Delta NO2_{ambient}}{\Delta O3_{ambient}} = -\frac{\Delta O3_{ambient}}{\Delta NO_{ambient}} = -1 \tag{6}$$





We define the Δ quantities as the change in concentration over 10 seconds. This step in the calibration procedure allows the sensitivity for NO to be expressed as a function of the $O_3$ sensitivity, as $\Delta NO_{ambient}$ is equal to $\Delta V_{NO}/k_{NO}$. Since $k_{NO2}$ is derived in Sect. 3.1, additional constraint for $NO_2$ sensor can be used as a consistency check.

Next we use these conservation equations to define the zero offsets for the NO and $O_3$ sensors. At nighttime, reaction R2 does not occur due to the absence of sunlight. In the absence of emissions, the NO concentration goes to zero on nights with sufficient $O_3$. Conversely, near strong emission sources, NO is found in excess of ozone and the $O_3$ concentration goes to zero (see Fig. 5). Using this logic, we identify times between 12 am to 3 am when there is zero NO or $O_3$ and define the zero offsets of the NO and $O_3$ sensors, using 1-minute averaged data with plumes excluded (see Sect. 3.3 for details of the plume
identification procedure).

The $NO_2$ offset can be determined using the pseudo-steady state (PSS) approximation. We estimate the $NO_2$ concentration through Eqn. 7:

$$j_{NO2}[NO_2] = k_{NO-O3}[NO][O_3] \tag{7}$$

Here, $j_{NO2}$ (in units of $s^{-1}$) is the photolysis rate constant for reaction R2 and $k_{NO-O3}$ (in units of $cm^3$ molecule$^{-1}$ $s^{-1}$) is the rate constant for reaction R1. $[X]$ expresses the concentration of gas $X$ in units of molecules $cm^{-3}$. We use calibrated, 1-minute average NO and $O_3$ concentrations measured from 12 pm to 3 pm with a time derivative of $O_3$ near zero to insure that the measurements reflect air that has achieved steady state. The $NO_2$ concentration at PSS is derived using Eqn. 7 and the $NO_2$ offset is chosen to insure the calculated and observed $NO_2$ is equal. $NO_2$ is also produced through the reaction of
$HO_2/RO_2$ with NO, but this is omitted from the right hand side of Eqn. 7, resulting in a lower bound of $NO_2$ concentration. Estimated $NO_2$ is therefore low by about 5% in winter and as much as 30% in summer.

**3.3 Use of co-emitted gases in plumes to calibrate the CO sensors' sensitivity**

The CO sensor cannot be constrained by cross sensitivity to the other gases. Instead, we constrain the sensitivity by insisting that the median emission factor of CO per unit $CO_2$ corresponds to median values reported for the U.S. vehicle fleet. We
express the CO emission factor ($EF_{CO}, ppb\ ppm^{-1}$) as in Eqn. 8:

$$EF_{CO} = \frac{\Delta CO_{ambient}}{\Delta CO2_{ambient}} = \frac{1}{k_{CO}} \frac{\Delta V_{CO}}{\Delta CO2_{ambient}} \tag{8}$$

Our measurements of the concentration of $CO_2$ are described in Shusterman et al. (2016) and values for $EF_{CO}$ are reported in Dallmann et al. (2013; see Table 2). We constrain the sensitivity of the CO sensors in the network such that the median $\Delta CO/\Delta CO_2$ of the plumes with high CO emissions are equal to emission factors characteristic of passenger vehicles.




Figure 6 shows an example of a measured plume and the derived CO emission factor. We identify plumes as the local maximum found in a 10-minute moving window, starting and ending at the local minima. Each plume is a few minutes in duration, representing an emission factor averaged over several vehicles. In the absence of $CO_2$ measurements, it would be possible to use the ratio of emission factors of $NO_x$ to CO, which is also widely measured (Mcdonald et al., 2013; Civerolo et al., 2016).

**3.4 Use of global background to calibrate the CO sensors' zero offset**

To infer the zero offset of the CO sensor, we follow the procedure outlined in Shusterman et al. (2016) for $CO_2$ sensors. We assume the signal measured at a given site is decomposed as in Eqn. 9:

$$[CO]_{node} = [CO]_{background} + [CO]_{local} + offset \tag{9}$$

The measurement of the pollutant $CO$ ($[CO]_{node}$) is the sum of regional and local signals ($[CO]_{background}$ and $[CO]_{local}$, respectively), as well as some offset from the true concentration ($offset$). Assuming the monthly minimum concentration measured at a given site represents $[CO]_{background}$, this background signal is compared to that measured at a "supersite" of reference instruments located within the network domain, allowing the offset to be derived. We also assume that when $[CO]_{node}$, as well as $[CO]_{local}$, is minimum in each day, the concentration measured at a given site has a constant deviation from the background signal. This is a reasonable assumption for the BEACO$_2$N domain as the dominant wind pattern frequently brings unpolluted air from the Pacific Ocean.

**3.5 Temperature dependence and temporal drift**

In order to account for the temperature dependence of calibration parameters, we apply the calibration process described in Sect. 3.1 through 3.4 for temperature increments of 1$^o$C to the data in 3-month running window. Then, we are able to define a temperature dependent sensitivity and zero offset, which is used to convert the measured voltages to mixing ratios. In this way, we can also evaluate temporal drift with monthly resolution. The calibration procedure can be repeated for shorter time intervals if wider temperature windows are used.

**4 Evaluation with reference observations**

We evaluate the efficacy of our calibration method using a BEACO$_2$N node co-located with reference instruments at the Laney College monitoring site maintained by the Bay Area Air Quality Management District (BAAQMD). Here we consider data collected from February to April 2016, calibrate it according to the procedure described above (following Sect. 3.1 to 3.5), and compare it against the BAAQMD data. Reference data is collected by a TECO 48i CO analyzer and a TECO 42i $NO_x$ analyzer. Ozone data from the "Oakland West" location, the closest ozone-monitoring site maintained by BAAQMD, was used for multiple linear regression of Eqn. 5. The zero offset for CO was calculated using BAAQMD data from the



Bodega Bay background site (see Fig. 1; Guha et al., 2016) as local "supersite" data was unavailable during this period. A background site closer to the network would likely improve our ability to constrain the CO zero offset; we installed a reference instrument for that purpose in summer 2017.

In our calibration procedure, the cross-sensitivities and temperature dependence is corrected for better accuracy. Table 3 shows the reduction in mean absolute error (MAE) that results when cross-sensitivity and temperature dependence issues are considered during multiple linear regression of Eqn. 7. Fully calibrated, hourly averaged BEACO$_2$N sensor data is compared to reference data in Fig. 7. For NO, NO$_2$, O$_3$, and CO the mixing ratio measured agrees reasonably well with the reference instrument ($r^2$ = 0.88, 0.58, 0.66, and 0.74 respectively) and is fairly accurate (MAE = 3.25 ppb, 4.00 ppb, 6.88 ppb, and

46.60 ppb respectively). The noise (±2σ) in the difference between calibrated BEACO$_2$N data and reference data is 10.47 ppb for NO, 10.08 ppb for NO$_2$, 13.81 ppb for O$_3$, and 111.04 ppb for CO. These noise values are dominated by the Alphasense noise except in the case of CO, where noise is evenly split between the low-cost sensors and reference instruments.

## 5 Examples of network performance

Figure 8 shows a week-long time series of fully calibrated air quality data from four BEACO$_2$N sites in 2017 (see Fig. 1).

BEACO$_2$N nodes capture the short-term variability associated with local emissions, superimposed on the diurnal variation caused by mixing and changes in the height of the boundary layer. Large mixing ratios of NO, NO$_2$, and O$_3$ are observed at the Hercules and Ohlone sites, representing strong NO$_x$ emissions from an oil refinery nearby. The spatial variability of trace gases observed at these 4 BEACO$_2$N sites provides a richer perspective on emissions when compared to that provided by the one regulatory monitoring site in the vicinity.

The emission ratios of CO and NO$_x$ were also investigated using the BEACO$_2$N data from sample locations. Figure 9 shows ratios observed at the Laney College site. The slope of CO/NO$_x$ varies from 2.43 to 18.12 across 5 BEACO$_2$N sites, reflecting spatial variations in local sources. Sites near roads with more diesel vehicles, such as Laney College, show lower CO/NO$_x$ ratios, as expected given diesel vehicles' higher NO$_x$ emissions. The range of observed CO/NO$_x$ emission ratios is

similar to the values reported by Mcdonald et al. (2013).

## 6 Conclusion

Calibration of low-cost sensors is necessary for quantitative analysis. In this paper, we have described a truly low cost, routine in-field calibration method and the subsequent evaluation of a fully calibrated low-cost, high-density air quality sensor network. The Alphasense B4 electrochemical gas sensors are able to detect typical diurnal cycles in gas

concentrations as well as short-term changes corresponding to chemical reactions and local emissions. These capabilities of





the sensors are utilized for a field calibration protocol that does not require co-location with reference instrumentation, but does require reference instruments to be sited within the network domain. The calibrated dataset demonstrates the accuracy required to resolve information relevant to urban emission sources, such as CO/NO$_x$ emission ratios. Through this work, we can realize the promise of low-cost, high-density sensor networks as a viable approach for atmospheric monitoring.

**Acknowledgements**

This work was funded by the Bay Area Air Quality Management District (2016.041), the Health Effects Institute (R-82811201), and the Koret Foundation. Additional support was provided by a Kwanjeong Educational Fellowship to Jinsol Kim, a NSF Graduate Research Fellowship to Alexis A. Shusterman, and a Hellman Family Graduate Fellowship to Kaitlyn J. Lieschke. We acknowledge the use of data sets maintained by BAAQMD's Ambient Air Monitoring Network, as well as
David M. Holstius, Holly L. Maness, and Virginia Teige for their contributions to BEACO$_2$N's code base.

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





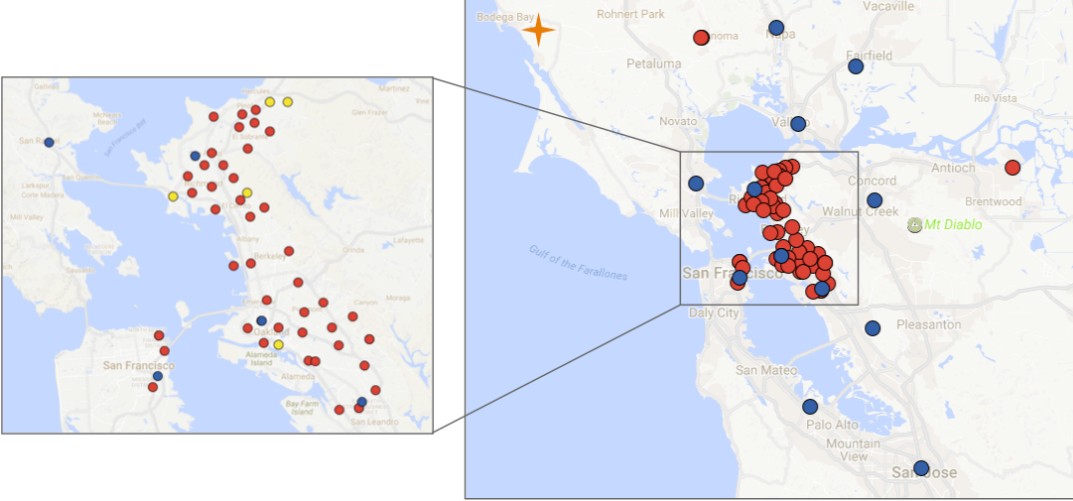

**Figure 1: Map of San Francisco Bay Area showing current BEACO₂N node sites (red), BAAQMD reference sites with O₃ measurements (blue), and the BAAQMD Bodega Bay regional greenhouse gas background site (orange). The sites used in this analysis are marked in yellow on the detailed panel.**

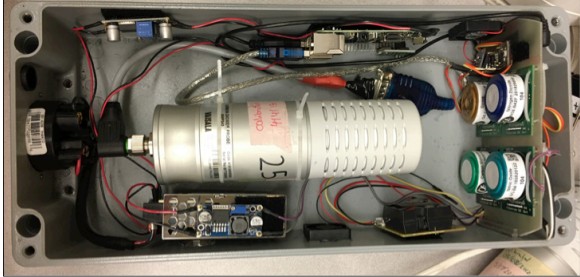

**Figure 2. Current BEACO₂N node design.**

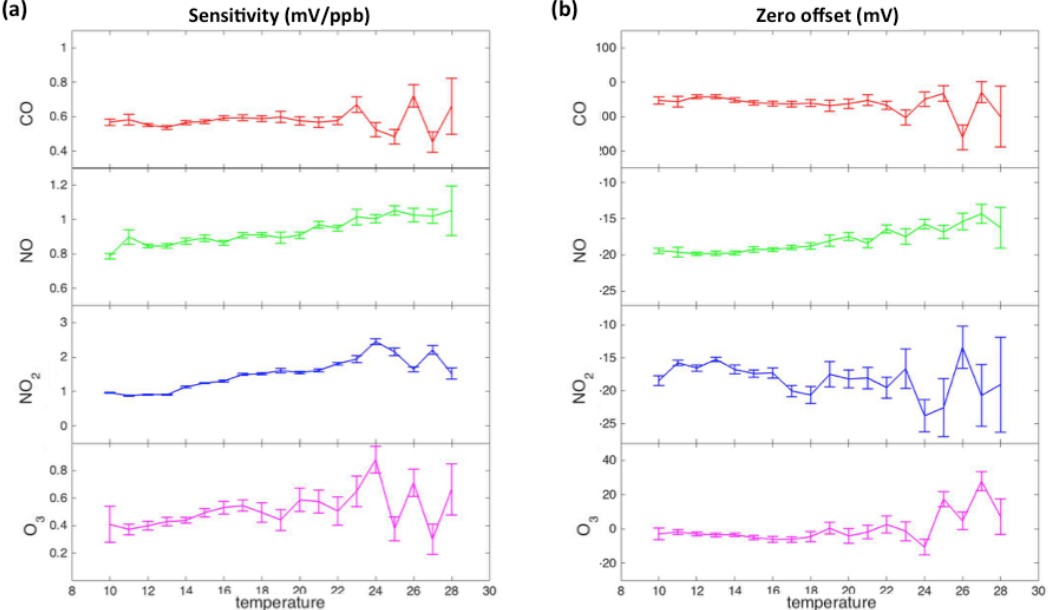

**Figure 3.** Representative temperature dependent sensitivities (a) and zero offsets (b) of the Alphasense electrochemical sensors calculated by comparing measurements from BEACO$_2$N node to measurements from reference instrument.

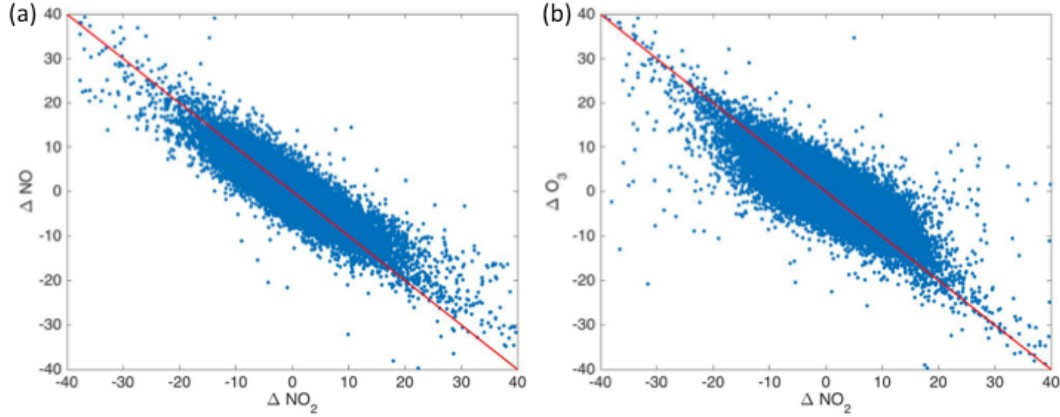

**Figure 4.** Correlation between (a) the derivative of NO and the derivative of NO$_2$ and (b) the derivative of O$_3$ and the derivative of NO$_2$ at 10 second resolution from a representative week of BEACO$_2$N data; red line has slope of -1.





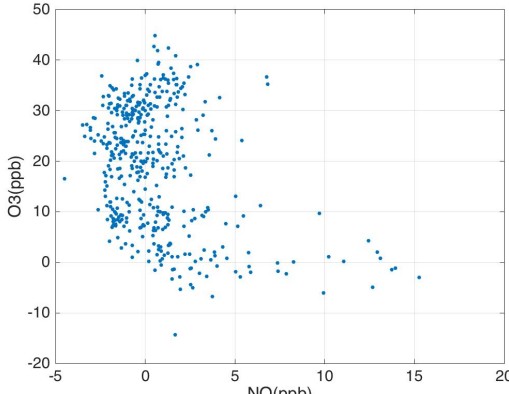

**Figure 5. Representative month of 1-minute averaged NO and O$_3$ measurements taken between 12 and 3 am; plumes excluded.**

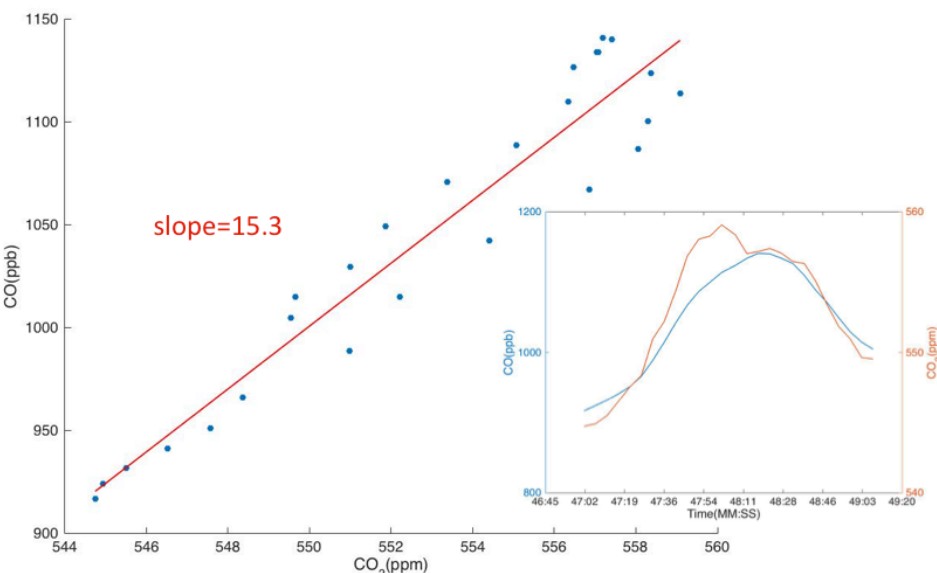

5    **Figure 6. Example of CO plume identification and regression against CO$_2$ to find the CO emission factor. The derived CO emission factor (CO/CO$_2$) for this example is 15.3 ppb ppm$^{-1}$.**



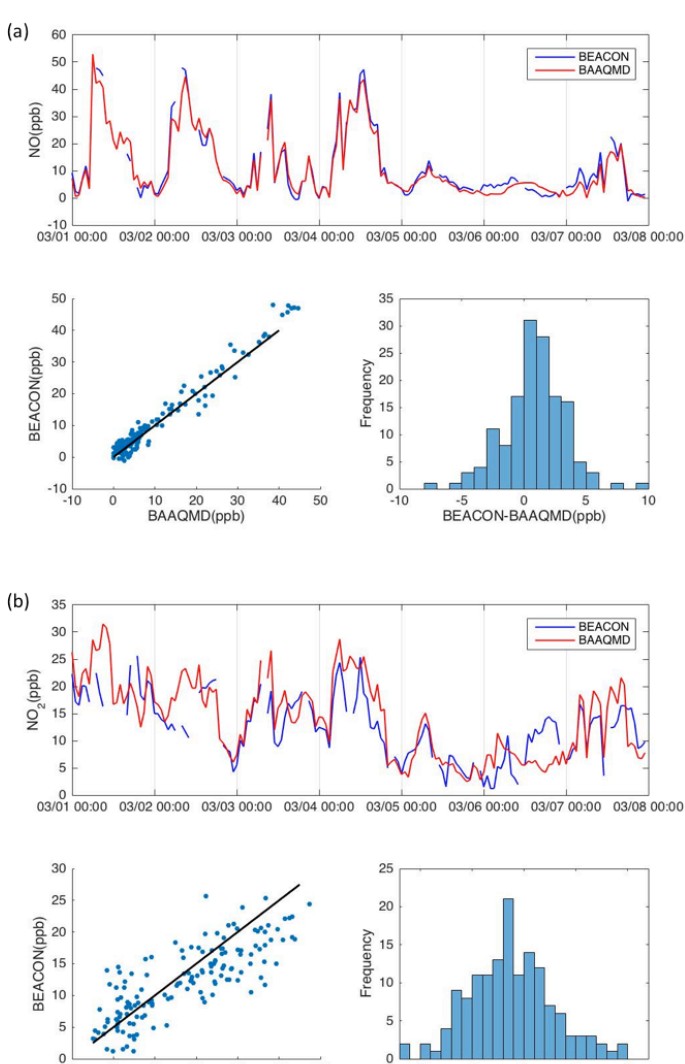





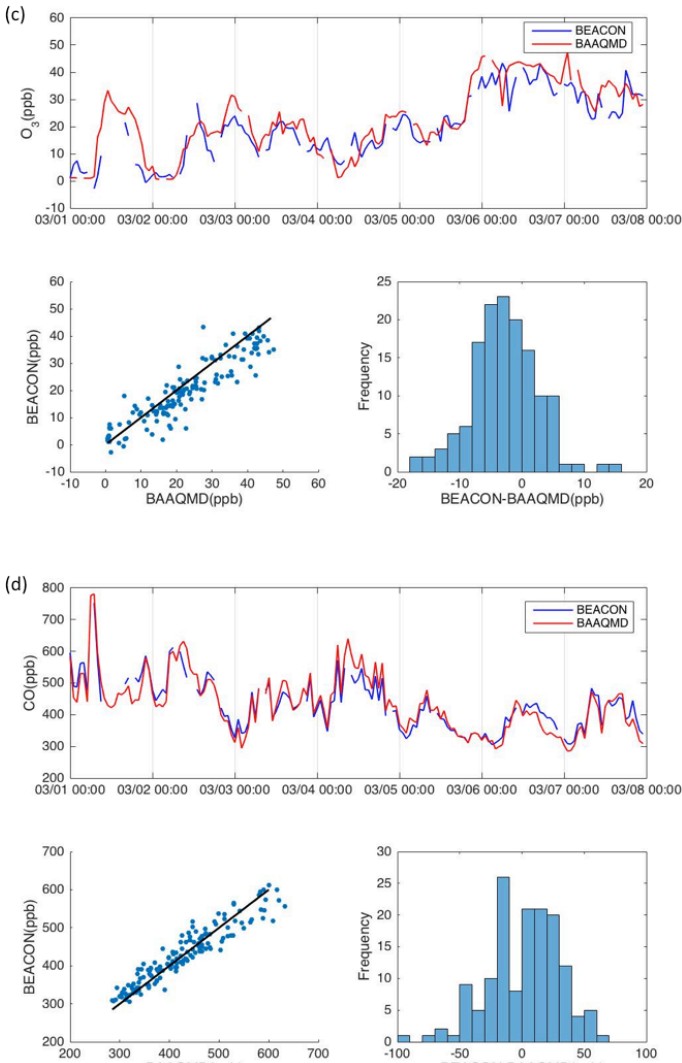

**Figure 7. Time series (top), direct comparison (bottom left), and histogram (bottom right) of hourly averaged (a) NO, (b) NO₂, (c) O₃, (d) CO mixing ratios from a representative week of BEACO₂N and BAAQMD data. Black line in bottom left plot indicates the**

5  **1:1 line.**



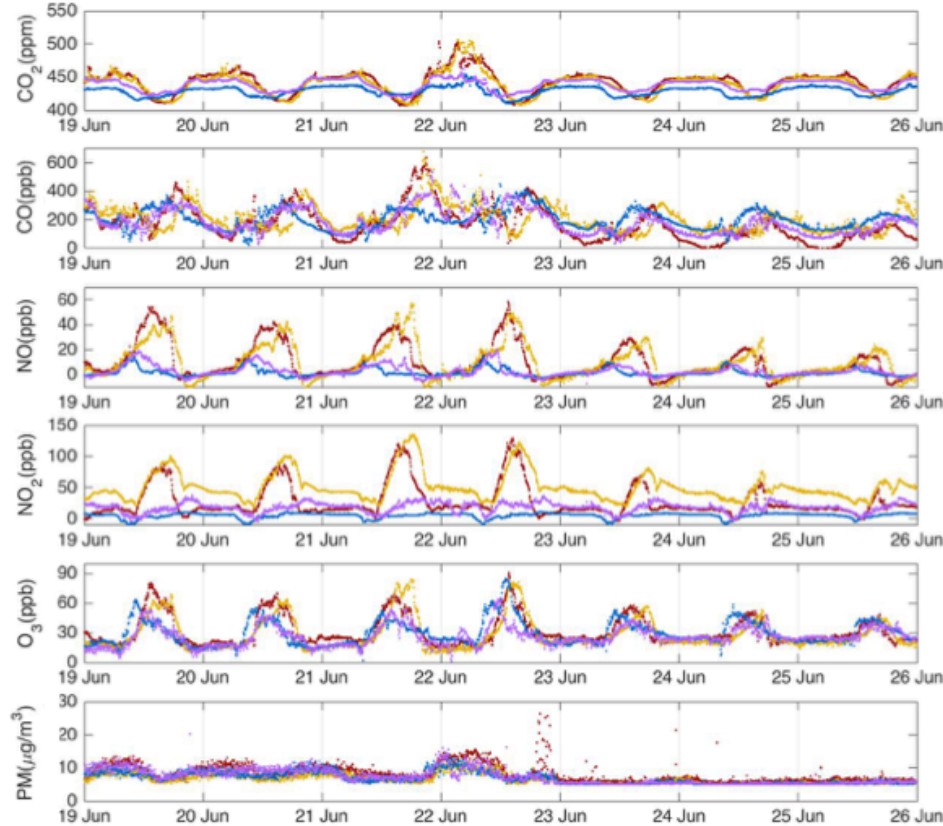

**Figure 8. Time series of fully calibrated BEACO₂N data from a representative week at 4 Richmond sites deployed in 2017. Observations from the Hercules, Ohlone, Washington, and Madera sites are plotted in red, yellow, blue, and purple, respectively. Particulate matter is converted to units of mass concentration according to Holstuis et al. (2014).**



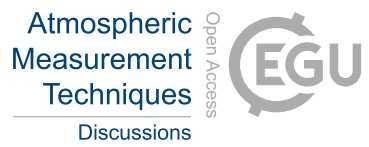

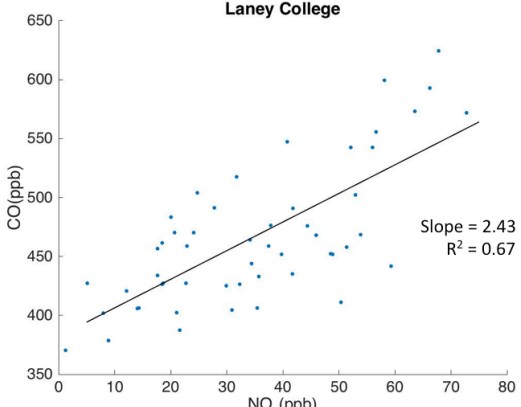

**Figure 9. CO vs. NO$_x$ measured between 8 am and 10 am.**



**Table 1. Zero offsets and sensitivities of a representative quartet of Alphasense B4 electrochemical sensors derived via comparison to delivered reference gases during two separate laboratory calibration separated by an approximately 10-week interlude.**

|  |  | May | August |
|---|---|---|---|
| $O_3$ | Zero offset (mV) | -34.6417 | -42.7629 |
|  | Sensitivity (mV/ppb) | 0.6404 | 0.2997 |
| CO | Zero offset (mV) | 108.9770 | 89.5812 |
|  | Sensitivity (mV/ppb) | 1.2192 | 1.0301 |
| NO | Zero offset (mV) | -14.2030 | -17.7801 |
|  | Sensitivity (mV/ppb) | 1.5758 | 1.2972 |
| $NO_2$ | Zero offset (mV) | -13.7159 | -6.0649 |
|  | Sensitivity (mV/ppb) | 0.4842 | 0.3843 |

**Table 2. Reported emission factors of diesel and gasoline vehicles (Dallmann et al., 2011; Dallmann et al., 2012; Dallmann et al.,**
5 **2013). Emissions from heavy-duty diesel trucks, which account for <1% of all vehicles, were removed to give the value for "adjusted" light-duty gasoline vehicles.**

| Vehicle Type | CO emission factor ($g\ kg^{-1}_{fuel}$) |
|---|---|
| Heavy-duty Diesel Truck | 8.0 ± 1.2 |
| Light-duty Gasoline Vehicles | 14.2 ± 0.12 |
| Adjusted Light-duty Gasoline Vehicles | 14.3 ± 0.7 |

**Table 3. Mean absolute error of comparison between regional $O_3$ and BEACO$_2$N $O_3$ measurements derived from multiple linear regression models of increasing complexity.**

| Regression Models | | Mean absolute error (ppb) |
|---|---|---|
| $O3_{true} = \dfrac{V_{O3}}{k_{O3}} - offset$ | Linearity of observed voltages and gas concentration | 14.4063 |
| $O3_{true} = \dfrac{V_{O3}}{k_{O3}} - \dfrac{V_{NO2}}{k_{NO2}} - offset$ | $O_3$ sensor's cross-sensitivity correction | 10.6795 |
| $O3_{true} = \dfrac{V_{O3}}{k_{O3}} - \dfrac{V_{NO2}}{k_{NO2}} + r_{NO-NO2}\dfrac{V_{NO}}{k_{NO}} - offset$ | $NO_2$ and $O_3$ sensor's cross-sensitivity correction | 8.8172 |
| $O3_{true} = \dfrac{V_{O3}}{k_{O3}} - \dfrac{V_{NO2}}{k_{NO2}} + r_{NO-NO2}\dfrac{V_{NO}}{k_{NO}} - offset$ | Adding temperature correction | 8.1360 |

