# Peer review of "The BErkeley Atmospheric CO2 Observation Network: Field Calibration and Evaluation of Low-cost Air Quality Sensors"

_Atmospheric Measurement Techniques, 2017_

## Referee Comment (RC1) · Anonymous Referee #1 · 25 Oct 2017

This article presents work from the deployment of low cost air quality sensors in high grid network around San Francisco Bay Area focusing mainly on alternative approach for field calibration of the low cost toxic gas sensors (CO, NO, NO2, O3) for some of the challenges described previously in literature. With the growing interest in the application of low cost sensors in air quality monitoring, the method presented here will add to the existing literature in this field. The manuscript is well written and the authors adequately describe their approach, validating the method by comparing to reference methods for the monitored gas species. I will like the authors to clarify a few points and some minor corrections outlined below.

[Figure]

Main comments

While most of the subsection in section 3 (Model for Field Calibration) are well presented, section 3.2 needs more clarification. What do the authors mean by "properly calibrated time derivative" in P6, line 24? Some of the description is not clear enough, lines 16-18.

As a general practice, I will like the authors to include the duration of the data used in generation the statistics and for some of the figures as this will allow the reader to put the result in context. For instance, Table 3, P21 shows the MAE of O3 without any information on the data period, none of these matches the 6.88 ppb MAE present for O3 in P9, line 9.

Can the authors explain why the O3 data shown in figure 8 appears to have a better noise < 11ppb ($2\sigma$) quoted for the lab tests? What are the temporal resolutions of the data presented in this figure? The reader will benefit if this information is included in figure caption or main text.

The authors need to clarify the VCO, VNO etc. in equations 1-4. Is this the voltage difference of the "working" and "auxiliary" electrodes or the just the "working" electrode.

Minor corrections

P.2, line 16, there is track change

P. 3, line 7: the Shusterman el al. reference is missing in the references.

P.4, lines 10-11: rewrite equations 3 and 4, suggest putting the cross interference terms (rNO-NO2 x NO ambient) in bracket.

P.9, line 7, this should read Eqn 5 not 7.

P14, add scale to figure 1, advise including image of deployed node in figure 2.

I suggest including the temperature plot in figure 8.

[Figure]

Several figures (Fig. 3, 4, 6 and 8) need to be replotted with legible axis labels.

A general comment, the authors should make sure numbers in chemical formulae are in subscript form.
* * *

---

## Referee Comment (RC2) · Anonymous Referee #2 · 22 Dec 2017

Review of: "The BErkeley Atmospheric CO2 Observation Network: Field Calibration and Evaluation of Low-cost Air Quality Sensors", by Jinsol Kim, et al.

This paper describes a novel approach to calibrate inexpensive sensor networks. The idea is to use known atmospheric chemistry relationships to constrain correlated measurements and derive corrections or calibrations. The paper is well presented and clear. The figures illustrate the main points well and support the conclusions of the paper. The paper, overall, is well suited to AMT and will make a valuable contribution to the growing area of sensor network research. I recommend publication after a few minor changes.

These issues are listed below. I believe that each of these can be addressed without major changes to the paper.

My overall impression with this approach is that if you know what the measurements should look like you can modify them to match this expectation. The case study does a good job of making this point. The manuscript does not address the alternative case that might not follow the expected chemical relations. The main question that I have after reading this paper is: How well does this approach work under less ideal circumstances? For example, the analysis assumes NOx + Ox is conserved, which is appropriate for being near a point source. How well does this approach work with a sensor that samples multiple sources where NOx + Ox is not conserved? Or, if CO/CO2 is different because of a large diesel presence. In other words, how useful is this approach in general? The answer to this question is a general point that needs to be developed better in the discussion.

Another aspect that should be discussed is the sensitivity of the calibrations to these assumptions. How large are these corrections, typically? If the NOx+Ox assumption is not correct by some amount, how does this impact your calibration? Likewise for CO/CO2.

---

## Author Comment (AC1) · 8 Feb 2018

**Response to Referee Comments:**

We thank the two referees for their detailed comments, which have been a great help to improve our manuscript.

In addition to our responses to the referee comments, further analysis has changed our understanding of how to use plumes in the calibration and led us to a substantial revision of Sections 3.2 and 3.3. We noticed that the behavior of the NO, NO2 and O3 sensors (see Eqn. 6 and Fig. 4), which we use to calibrate NO sensors' sensitivity, is caused by the cross-sensitivity of the sensors (see Eqn. 1-4) and not the chemical conservation equations as we had previously thought (see Reactions 1-3). An alternative constraint is proposed in the revised manuscript and the reported values for accuracy evaluation as well as the figures in Sections 4 and 5 have changed slightly. Also, analysis of the recently released NO2-B43F and Ox-B431 sensors are now included in the manuscript. The basic approach and overall message of the paper are unchanged.

**Referee #1 Comments: (Referee comments in italics)**

This article presents work from the deployment of low cost air quality sensors in high grid network around San Francisco Bay Area focusing mainly on alternative approach for field calibration of the low cost toxic gas sensors (CO, NO, NO2, O3) for some of the challenges described previously in literature. With the growing interest in the application of low cost sensors in air quality monitoring, the method presented here will add to the existing literature in this field. The manuscript is well written and the authors adequately describe their approach, validating the method by comparing to reference methods for the monitored gas species. I will like the authors to clarify a few points and some minor corrections outlined below. C1 AMTD Interactive comment Printer-friendly version Discussion paper

1) While most of the subsection in section 3 (Model for Field Calibration) are well presented, section 3.2 needs more clarification. What do the authors mean by "properly calibrated time derivative" in P6, line 24? Some of the description is not clear enough, lines 16-18.

P6, line 24 has been deleted due to the revision mentioned at the beginning of this response. We have updated the text to clarify P7, lines 16-18:

"We use sensitivity corrected (see Section 3.1 and 3.2), 1-minute average NO and  $O_3$  concentrations measured from 12 pm to 3 pm, and select data with a time derivative of  $O_3$  near zero to insure that the measurements reflect air that has achieved steady state."

2) As a general practice, I will like the authors to include the duration of the data used in generation the statistics and for some of the figures as this will allow the reader to put the result in context. For instance, Table 3, P21 shows the MAE of O3 without any information on the data period, none of these matches the 6.88 ppb MAE present for O3 in P9, line 9.

The analysis of the Laney College monitoring site used data from February to April 2016 as mentioned in P8, line 25-26 (P8, line 26-17 in revised manuscript). MAE values in Table 3 are calculated after conducting the multiple linear regressions explained in Section 3.1, and MAE in P9, line 9 is calculated after fully calibrating the data following the procedure from Sections 3.1 to 3.5, causing the difference in reported MAE values. We have added the following text for clarification:

"Here, MAE is calculated after conducting the sensitivity correction explained in Section 3.1, but before the offset correction in Section 3.3"

3) Can the authors explain why the O3 data shown in figure 8 appears to have a better noise < 11ppb ( $2\sigma$ ) quoted for the lab tests? What are the temporal resolutions of the data presented in this figure? The reader will benefit if this information is included in figure caption or main text.

The  $O_3$  data shown in Figure 8 is hourly averaged data, and the noise quoted from the laboratory tests is calculated from 10 s resolution data. We have added missing information about the resolution and period of the data in the figure captions and the main text.

4) The authors need to clarify the VCO, VNO etc. in equations 1-4. Is this the voltage difference of the "working" and "auxiliary" electrodes or the just the "working" electrode.

We have updated the text:

"Here, CO, NO, NO2, and O3 with the subscript "ambient" refer to the gas mixing ratios (ppb) in air;  $V_{CO}$ ,  $V_{NO}$ ,  $V_{NO_2}$  and  $V_{O_3}$  are the signals (mV) measured by each sensor, which is the voltage of the auxiliary electrode subtracted from the voltage of the working electrode; [...]"

**Minor Comments:**

1) P.2, line 16, there is track change

We have deleted the track change from the text

2) P.3, line 7: the Shusterman et al. reference is missing in the references.

We have added a reference to Shusterman et al. in the References.

3) P.4, lines 10-11: rewrite equations 3 and 4, suggest putting the cross interference terms (rNO-NO2 x NO ambient) in bracket.

We have rewritten Equations 3 and 4 as suggested.

4) P.9, line 7, this should read Eqn 5 not 7.

We have updated the numbering of the equations.

5) P14, add scale to figure 1, advise including image of deployed node in figure 2.

We have added scales to Figure 1 and included an image of a deployed node in figure 2.

6) I suggest including the temperature plot in figure 8.

We have added a temperature plot to Figure 8.

7) Several figures (Fig. 3, 4, 6 and 8) need to be replotted with legible axis labels.

We have re-plotted all of the Figures in the interest of legibility.

8) A general comment, the authors should make sure numbers in chemical formulae are in subscript form.

We have updated the chemical formulae to ensure that numbers are in subscript form.

**Referee #2 Comments: (Referee comments in italics)**

This paper describes a novel approach to calibrate inexpensive sensor networks. The idea is to use known atmospheric chemistry relationships to constrain correlated measurements and derive corrections or calibrations. The paper is well presented and clear. The figures illustrate the main points well and support the conclusions of the paper. The paper, overall, is well suited to AMT and will make a valuable contribution to the growing area of sensor network research. I recommend publication after a few minor changes. These issues are listed below. I believe that each of these can be addressed without major changes to the paper.

1) My overall impression with this approach is that if you know what the measurements should look like you can modify them to match this expectation. The case study does a good job of making this point. The manuscript does not address the

alternative case that might not follow the expected chemical relations. The main question that I have after reading this paper is: How well does this approach work under less ideal circumstances? For example, the analysis assumes NOx + Ox is conserved, which is appropriate for being near a point source. How well does this approach work with a sensor that samples multiple sources where NOx + Ox is not conserved? Or, if CO/CO2 is different because of a large diesel presence. In other words, how useful is this approach in general? The answer to this question is a general point that needs to be developed better in the discussion.

The reviewer raises a good question about how effective our calibration approach will be under a range of ambient conditions. While we are also interested in that question, providing a thorough answer is beyond the capabilities of our existing data set. As for the San Francisco Bay Area, hourly traffic data obtained from the Caltrans Performance Measurement System (PeMS) shows that diesel trucks typically account for < 10% of the total freeway traffic in the BEACO2N domain, with relatively little (~3%) intra-domain variation in the diesel truck fraction. Thus, we do not expect large variability in the CO/CO2 and NOx/CO2 ratio in our domain. We have added the text:

"Since diesel trucks have an order of magnitude higher NOx emission factors compared to gasoline vehicles, the percentage of truck traffic near each site affects the median emission factors. The median freeway truck ratio varies little across the BEACO2N network, however, regions with a larger range of median truck ratios will have larger uncertainties or require a calibration approach that accounts for this variation."

Conservation of NOx +Ox is now not used as an assumption due to our revised approach to calibrating sensors.

2) Another aspect that should be discussed is the sensitivity of the calibrations to these assumptions. How large are these corrections, typically? If the NOx+Ox assumption is not correct by some amount, how does this impact your calibration? Likewise for CO/CO2.

An example of sensitivities and zero offsets for calibration are shown in Figure 3. Our assumptions are directly constraining the concentration values. In other words, if there is 10% change in the constraining value, concentration of calibrated data will show 10% change. Conservation of NOx +Ox is now not used as an assumption due to our revised approach to calibrating sensors.

[revised manuscript text omitted]

---

## Referee Report (RR1)

Review of Revision 1: "The BErkeley Atmospheric CO2 Observation Network: Field Calibration and Evaluation of Low-cost Air Quality Sensors", by Jinsol Kim, et al.

This revision adequately addresses my comments and questions.  The changes to the paper make the paper better and publishable.